# Tropical fishes vanished after the operation of a nuclear power plant was suspended in the Sea of Japan

**Reiji Masuda** [ID]*

Maizuru Fisheries Research Station, Field Science Education and Research Center, Kyoto University, Maizuru, Kyoto, Japan

* reiji@kais.kyoto-u.ac.jp

**Data Availability Statement:** All relevant data are within the manuscript and its Supporting Information file.

## Abstract

Thermal discharge from a nuclear power plant (NPP) provides an opportunity to foresee changes in faunal communities that may be induced by ocean warming. I assessed these changes by identifying characteristics of the fish community near the thermal discharge from a NPP and by recording temporal changes that occurred after the suspension of the NPP. Underwater visual censuses were conducted near Takahama NPP in the Sea of Japan, and fish assemblages were compared to those in two other sites: a site with discharge from a coal-fired power plant and a control site. During the surveyed period (8 years) when the NPP was in operation, the sea water temperature at the site near the NPP was warmer, had a significantly higher fish abundance, and a higher species richness, including tropical fishes, than the other two sites. However, once the NPP was suspended, tropical fishes dramatically decreased near the NPP. This abrupt change in fish assemblage may be due to the lowest lethal temperatures of tropical fishes being only slightly higher than the winter temperature in this area. Relatively poor ecosystem structure in the local warming area may also have contributed to low resilience of tropical fish species to this temperature change.

## Introduction

Climate change and ocean warming are major concerns from both an ecological and economical perspective [1, 2]. Reported impacts of warming trends in the ocean are ubiquitous in both tropical and temperate waters. Projected negative impacts of oceanic warming are, however, most serious in tropical waters due to the loss of coral (i.e., as a result of coral bleaching) and subsequent habitat loss [3]. Changes are also conspicuous in temperate waters with an increased number of warm water-associated fish species or tropical vagrants [4–6].

The Japanese Archipelago is suitable for the study of distributional change of organisms due to ocean warming. Although larvae and early settlers are consistently supplied from Okinawa and other southern districts to temperate waters via the Kuroshio and Tsuhima currents (Fig 1), overwintering of tropical vagrants is limited by the lowest winter temperature [7]. The

**Funding:** This study was partly supported by the CREST program from the Japan Science and Technology Agency (grant number: JPMJCR13A2; http://www.jst.go.jp/kisoken/crest/en/project/33/e33_13.html) and JSPS KAKENHI Grant Number 19H05641. The funders had no role in study design, data collection and analysis, decision to publish, or preparation of the manuscript. There was no additional funding received for this study.

**Competing interests:** The authors have declared that no competing interests exist.

Sea of Japan is a hotspot of ocean warming, with the central area having experienced a warming speed of 1.70°C per century. This is much faster than the global mean (0.53°C) or the overall mean around Japan (1.09°C) over the same time period (Data from Japan Meteorological Agency, http://www.data.jma.go.jp/gmd/kaiyou/data/shindan/a_1/japan_warm/japan_warm.html, accessed on February 27, 2018). The warming trend is projected to either have a positive effect (e.g. extension in the distribution of the commercially important predatory fish yellowtail *Seriola quinquradiata* [8]), or a negative effect (e.g. the loss of seaweed vegetation [9]). Projected winter sea surface temperatures in the Sea of Japan in 2050 and 2100 will be approximately 2°C and 4°C higher than the present level, respectively, under the Intergovernmental Panel on Climate Change A1B scenario (medium scenario of greenhouse gas emissions) [8].

A nuclear power plant (NPP) consistently requires water to cool reactors during operation. All the NPPs in Japan are located along the coastline and discharge water into the ocean that is warmer than that of the natural environment [10] (Fig 1). Although the impacts of thermal discharge have been reported to be substantial in freshwater fish communities [11], the impact on marine ecosystems is controversial [12–14]. For example, a major decrease in algae and an increase in invertebrate grazers were reported with thermal discharge in California [12], however, no measurable change in fish assemblage was detected in response to the operation of a NPP in Taiwan [13]. In addition, Hillebrand et al. [15] reported an increased turnover of the algal community in the thermal discharge of a NPP in the Bothnian Sea. Local warming induced by thermal effluent may provide a unique opportunity to foresee changes that may occur in near-future faunal communities in temperate waters as a result of climate change.

The primary goal of the present study was to determine the effect of thermal discharge on the surrounding temperate fish community as a proxy for understanding the impact of future temperature increases in the region. Underwater visual surveys were conducted at the site of thermal discharge from a NPP and the observed fish assemblage was compared with those in reference sites. Preliminary surveys revealed that sea water temperature in the NPP site was approximately 2°C warmer than in the reference sites, and thus the fish assemblages under the influence of the NPP may be representative of those under the 2050 projection in other parts of the region. This study aimed to assess: 1) if thermal discharge supported a distinct fish community in the local warming area, such as a higher number of tropical or warm-water origin species in winter; and 2) if such a fish community would collapse following the suspension of the NPP. Surveys were conducted in winter because overwintering individuals, rather than early settlers in summer or autumn, are likely to have more impact on the local ecosystem through actions such as grazing on algae and competing for resources with temperate species.

## Methods

### Ethics statement

Underwater visual surveys were conducted in accordance with local and governmental laws and regulations. Research was performed according to the guidelines of the Regulation on Animal Experimentation at Kyoto University. Field research in Nagahama was approved by the harbormaster of Maizuru Bay on a quarterly basis, whereas no permissions were required for the other two sites. The field study did not involve endangered or protected species. No fish or other animals were sacrificed for the purpose of this study.

### Study sites, census method and data analyses

Underwater visual censuses were conducted at three sites: Otomi (35°32'N, 135°30'E), Sezaki (35°33'N, 135°21'E), and Nagahama (35°29'N, 135°22'E), along the coast of Wakasa Bay, Sea

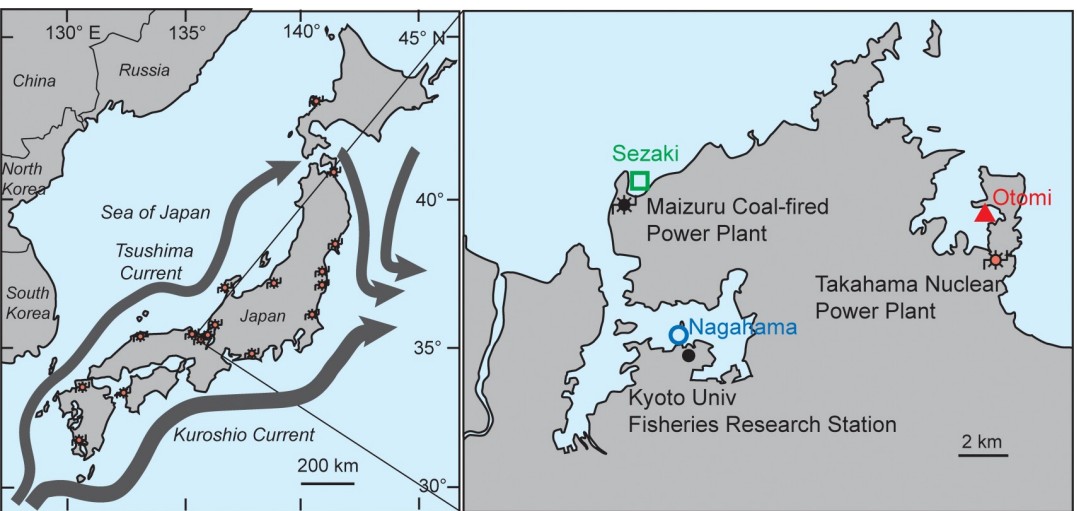

**Fig 1. Map depicting the sites (Otomi, Sezaki, and Nagahama) where the visual censuses were conducted along the coast of Wakasa Bay, Sea of Japan.** Red symbols represent the locations of nuclear power plants, while the green symbol represents the site with a coal-fired plant and the blue symbol represents the control site.

of Japan (Fig 1). Visual censuses began in January 2004 and were conducted four times every winter (from late January to early March) over the subsequent 14 years in Otomi and 11 years in Sezaki (S1 Table). The Otomi site was located 2 km from the exhaust of the thermal discharge of the Takahama NPP. Takahama NPP had four reactors, the first began operating in 1974 and the last reactor began operating in 1984, with a total discharging rate of 238 m$^3$ s$^{-1}$ thermal effluent that was 7°C higher in temperature than that of the natural environment [10]. The operation was suspended from February 20, 2012 to May 2017 due to the accident at the Fukushima NPP in March 2011. The census point in Sezaki was adjacent to the thermal discharge of the Maizuru coal-fired power plant (CPP), which was in operation from August 2004; therefore, the effect of the thermal discharge was expected to be evident from the winter of 2005 or thereafter. The maximum amount of discharge was 73 m$^3$ s$^{-1}$. (https://www.city.maizuru.kyoto.jp/kurashi/cmsfiles/contents/0000000/545/5_karyoku(hp).pdf, accessed on May 24, 2019). Nagahama served as a control site where censuses have been conducted twice every month since January 2002 [4, 16]. Winter data from this site were used for comparison.

Areas of approximately 1200 m$^2$ (2 m in width × 600 m in length) were surveyed in each census. A modified line transect method termed "fin-kick transect" was employed, in which distance traveled was estimated by the number of fin kicks made [17]. This way the area surveyed slightly changed in each survey trial except around the starting point. The size of survey areas was decided to maximize detection of rare species within limited diving time based on previous studies conducted at the Nagahama and Otomi sites for other purposes [4, 18]. The relatively narrow width of the area allowed for consistency of detection under fluctuating visibility, and the low fish density in cold temperatures required this survey area. The species, estimated body length, and number of individuals of all the fish encountered along the transects were recorded. Fish species were identified according to Nakabo [19]. Surveys were conducted when the visibility was at least 1 m, but it was usually 3–8 m. Each survey took approximately 60 min. Mean (± SD) depth was 3.8 ± 1.2, 9.1 ± 3.8, and 3.4 ± 2.6 m in the Otomi, Sezaki, and Nagahama sites, respectively. Bottom water temperature was measured at the deepest point of the surveyed sites (5, 14, and 9 m in the Otomi, Sezaki, and Nagahama sites, respectively)

using an alcohol stem thermometer that was occasionally calibrated by a mercury standard thermometer.

The center of distribution (COD), defined as the mean southern and northern latitudinal limit of distribution in the northern hemisphere, was calculated using Nakabo [19] for all recorded species. The mean of COD in each survey was then used as a criteria of spatio-temporal tropicalization.

Statistical analyses were conducted using R 3.5.2 [20], with the packages 'lme4' [21], 'arm' [22] and 'vegan' [23]. Linear mixed models were fitted to each of bottom water temperature, fish species richness, fish abundance (fourth-root transformed to improve homoscedasticity) and COD as response variables, survey sites as fixed effects, and survey years with four replicate surveys nested below as random effects. The analyses were conducted separately in two periods when NPP was in operation (from 2004 to early February 2012) or suspended (February 20, 2012 and later). Fish assemblages were also analyzed by categorizing them into tropical, subtropical, and temperate species based on FishBase [24]. Fish abundance and species richness in each site, pre- and post-suspension of the NPP and the above category (tropical, subtropical and temperate species) were compared using linear mixed models. Confidence intervals of 95% were calculated for each variable by simulating linear mixed model fits (10000 runs). Differences between variables were considered significant ($p < 0.01$) if confidence intervals did not overlap [25]. This was applicable to comparisons between sites within a period (NPP on or off), as well as between periods within a site.

Fish assemblages were further assessed by non-metric multidimensional scaling (nMDS) [26]. Abundance data were summed for each species in each year and fourth-root transformed. Fish assemblages were then compared among sites and years using the Bray–Curtis similarity index. The differences between two periods (up to 2012 vs 2013–2017) were compared in each site by PERMANOVA.

## Results

The bottom water temperature dropped from 11.8˚C to 10.4˚C in the Otomi site from February 14 to 22, 2012, the Takahama NPP being suspended between these two surveys (February 20) (S1 Table). The water temperature remained consistently low throughout the remainder of the study period (Fig 2A). No major changes in bottom water temperature were detected in the Nagahama or Sezaki sites throughout the research period. The mean bottom water temperature in the Otomi, Sezaki and Nagahama sites before and after the suspension of the Takahama NPP were 13.6, 11.7, and 12.3˚C, and 10.6, 11.0, and 11.8˚C, respectively (Fig 3A). Temporal change of temperature was significant only in the Otomi site ($p < 0.01$). Temperature in the Otomi site was significantly higher than the other two sites when the NPP was on, whereas it dropped to lower than the Nagahama site when NPP was off probably due to the shallower bottom depth.

The total fish abundances found throughout the study period were 13683, 7250, and 4074, and the number of fish species were 65, 58, and 48 in the Otomi, Sezaki and Nagahama sites, respectively (S1 Table). Both the fish abundance and species richness decreased in the Otomi site after the suspension of the Takahama NPP, whereas such a change was not detected in the other two sites (Fig 2B and 2C). The COD in the Otomi site increased after the suspension of the Takahama NPP, but not in the other two sites (Fig 2D). Only the Otomi site showed a significant temporal change in each of the above variables represented by the lack of overlap in confidence intervals ($p < 0.01$, Fig 3B, 3C and 3D). The variables in the Otomi site were significantly different from the other two sites when the NPP was in operation ($p < 0.01$), while the difference was not detected when the NPP operation was suspended.

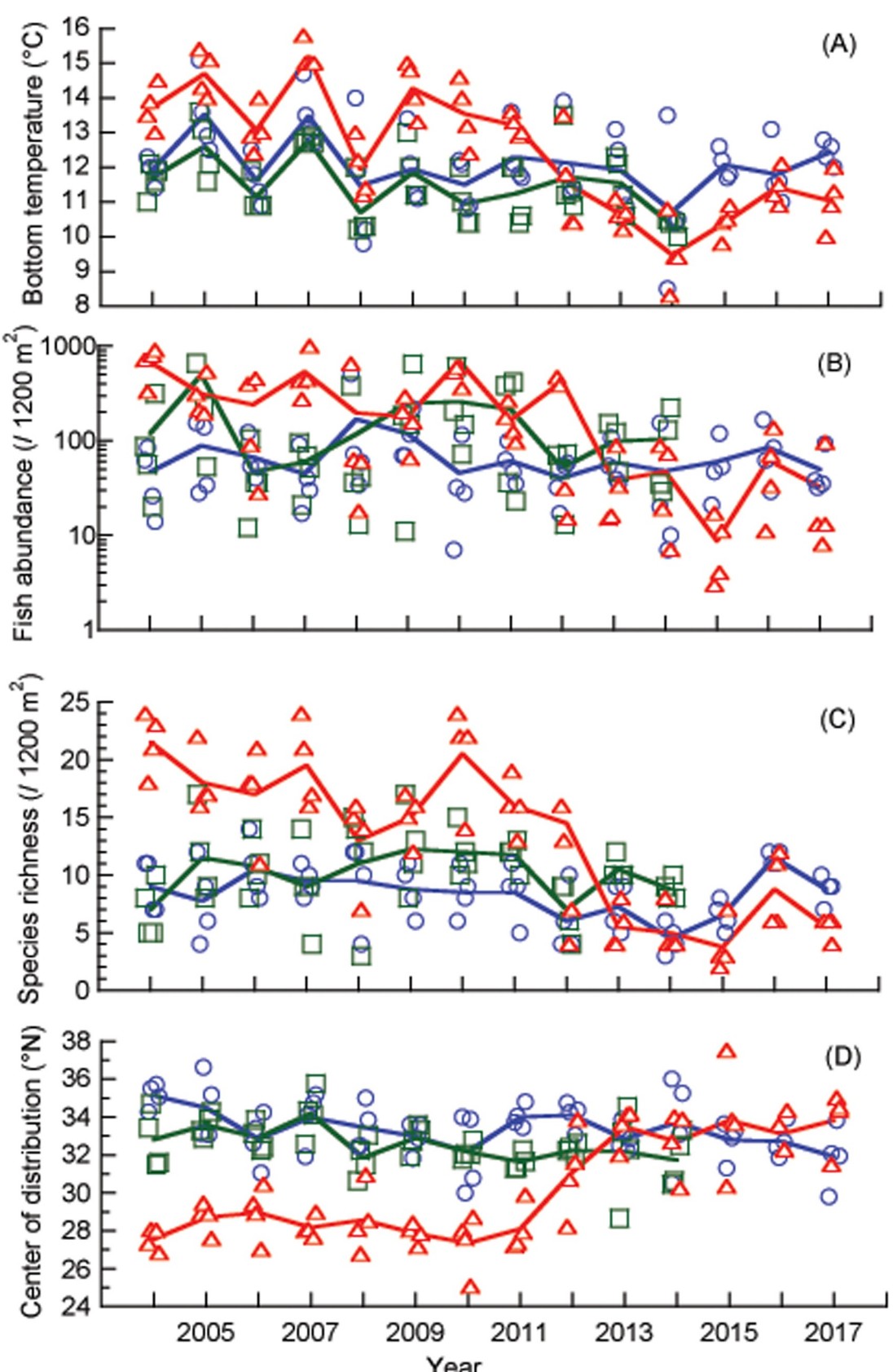

**Fig 2.** Water temperature (A), fish abundance (B), species richness (C) and mean value of the center of distribution in the northern hemisphere (D) in the three surveyed sites: Otomi (red triangles) Sezaki (green squares) and Nagahama (blue circles). Lines represent yearly means for each site (surveys = 4).

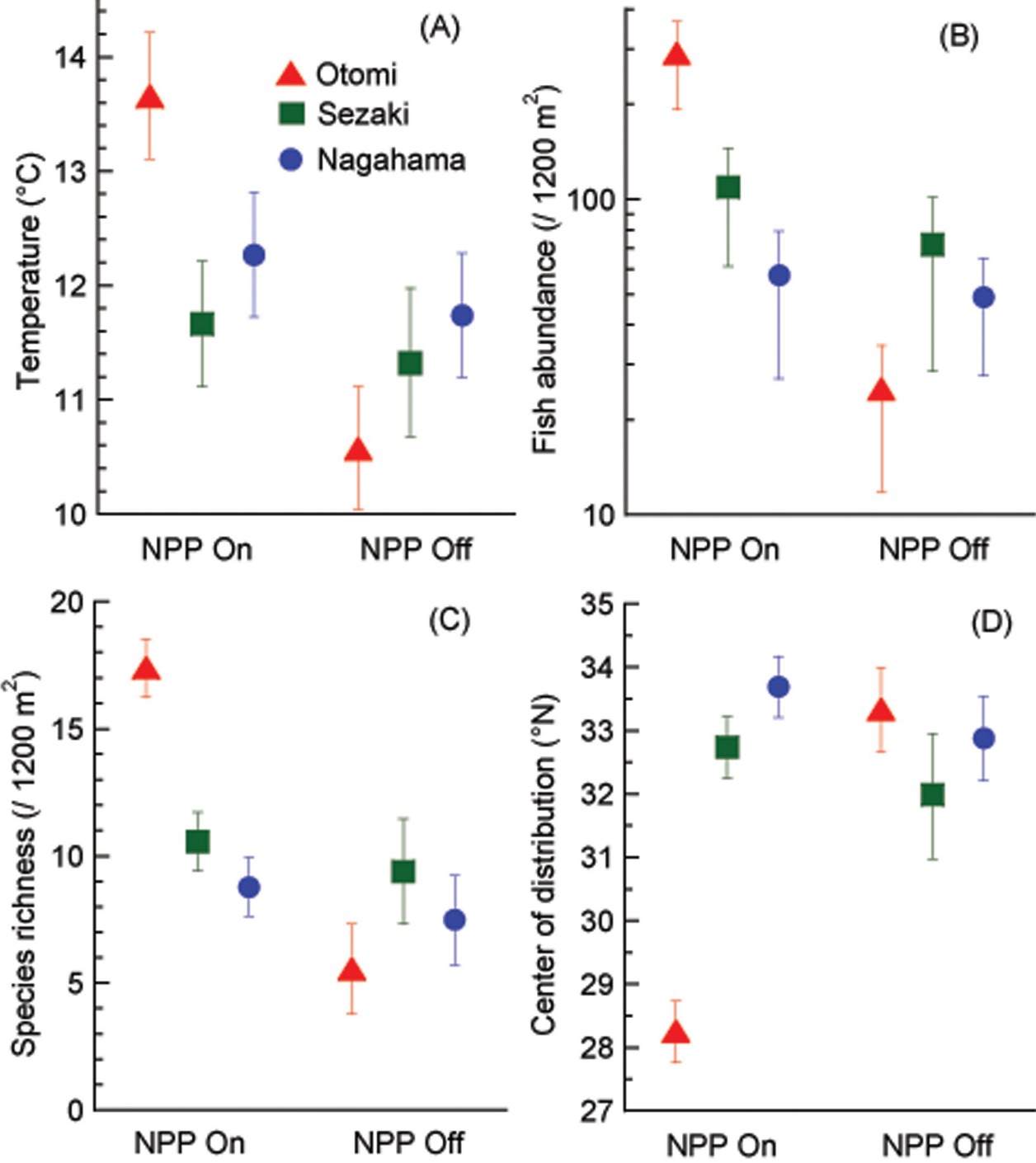

**Fig 3.** Within-period means when the Takahama nuclear power plant was in operation (NPP On: up to February 14, 2012) and when its operation was suspended (NPP Off: February 20, 2012 and later) for water temperature (A), fish abundance (B), species richness (C) and mean value of the center of distribution in the northern hemisphere (D) in the three survey sites. Error bars represent 95% confidence intervals.

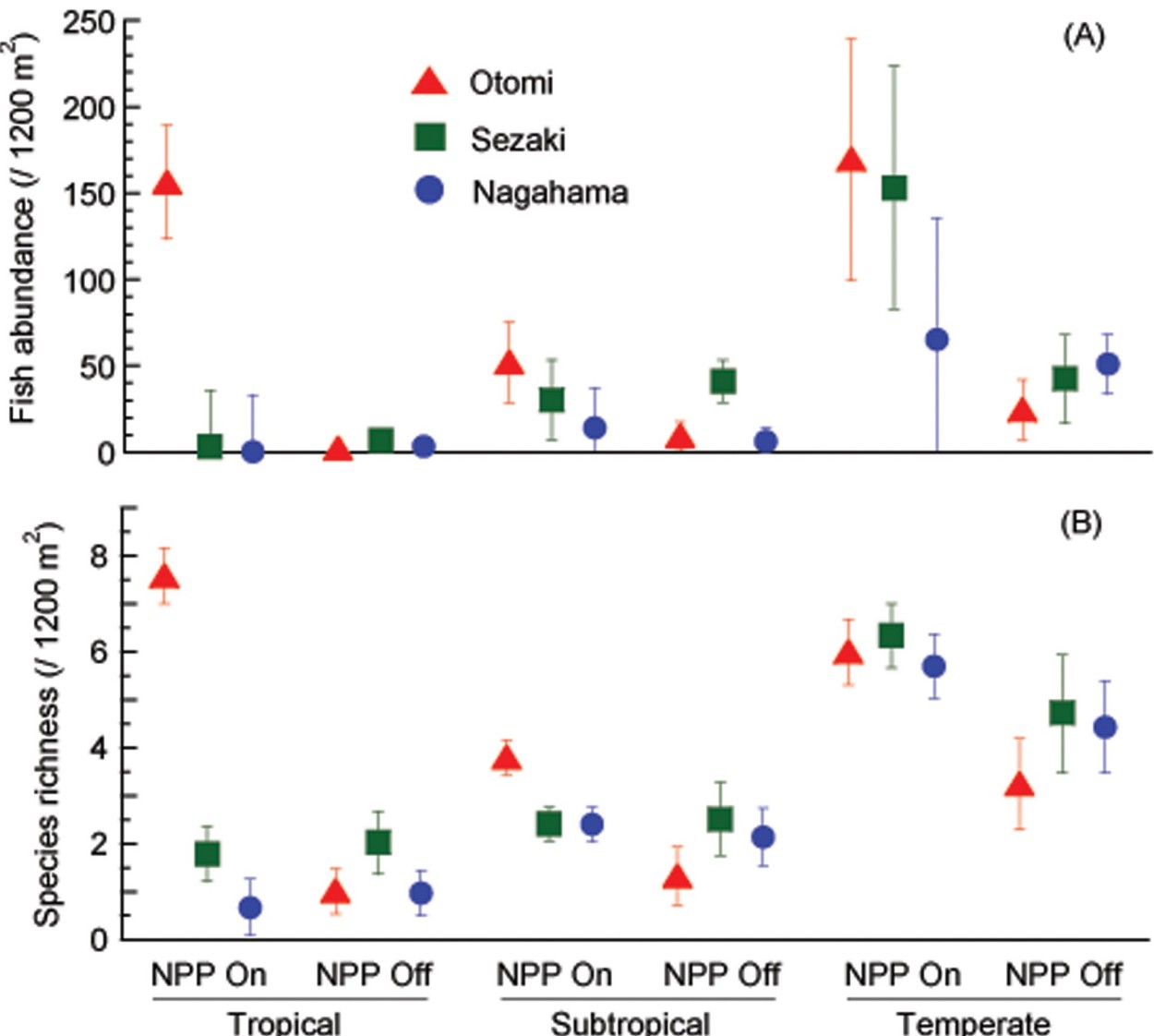

**Fig 4. Mean temporal changes in fish species composition following the suspension of thermal discharge.** Fishes were categorized into tropical, subtropical, and temperate species according to FishBase [24]. A: abundance, B: species richness. Error bars represent 95% confidence intervals.

Regarding the changes in the tropical, subtropical, and temperate species categories, tropical fishes dramatically decreased with the suspension of NPP in Otomi (Fig 4, $p < 0.01$ comparing pre- and post-suspension of NPP). The decrease was also significant in subtropical and temperate fishes in this site ($p < 0.01$). In Sezaki and Nagahama, the composition of fish species remained stable between these two periods, except for the significant decrease in temperate fish abundance ($p < 0.01$) in Sezaki (Fig 4A). Inter-site comparisons revealed that the Otomi site had significantly higher tropical fish abundance and species richness and subtropical fish species richness than the other two sites when the NPP was in operation.

The plotting of nMDS revealed that fish assemblages were distinct among the four groups (i.e. the Otomi site from 2004 to 2012, the Otomi site from 2013 to 2017, Sezaki site, and Nagahama site; stress level = 0.15, Fig 5). Fish assemblages in the Otomi site showed a significant change between 2004 to 2012 and 2013 to 2017 ($p = 0.001$, PERMANOVA), whereas no

significant change was detected in the Sezaki ($p = 0.085$) and Nagahama ($p = 0.095$) sites between these periods.

The most dominant fish species in the Otomi site prior to the suspension of the NPP were Japanese anchovy *Engraulis japonicus*, the cobalt silverside *Hypoatherina tsurugae*, and half-line cardinal *Apogon semilineatus*, representing 16.8, 15.5 and 12.9% in total abundance up to the suspension of the NPP, respectively. After the suspension, the banded goby *Pterogobius elapoides* and pearl-spot chromis *Chromis notatus notatus* were relatively abundant (52.7 and 23.0%, respectively). In Sezaki, *E. japonicus* was the most dominant species, followed by *C. notatus notatus*, Japanese sand lance *Ammodytes personatus* and the bambooleaf wrasse *Pseudolabrus sieboldi* throughout the survey (43.7, 9.9, 9.7 and 9.3%, respectively). In Nagahama, the chameleon goby *Tridentiger trigonocephalus*, followed by the *E. japonicus*, striped sand goby *Acentrogobius virgatulus*, and black rockfish *Sebastes cheni* dominated throughout the survey (34.2, 30.7, 8.3 and 5.8%, respectively).

Two tropical fish species, the blue damselfish *Pomacentrus coelestis* and cutribbon wrasse *Stethojulis interrupta*, were consistently recorded in the Otomi site during the operation of the Takahama NPP (Fig 6A and 6B; S1 Table). However, neither of these species were recorded after its suspension. Die off of warm water fish species, such as *H. tsurugae* and the flatfish *Pseudorhombus pentophthalmus*, as well as invertebrates, such as the long-spined sea urchin *Diadema* sp. and ashy sea cucumber *Holothuria pervicax*, were also observed in two surveys after the suspension (Fig 6C and 6D). These tropical echinoderms were common until 2012, but were not observed during the surveys in 2013 and thereafter.

Little to no vegetation was observed in the Otomi site (Fig 6A and 6B), however, vegetation such as *Sargassum* and other brown and red algae were observed in the other two sites. Herbivorous rabbitfish *Siganus fuscescens* were abundantly recorded only in the Otomi site while the NPP was in operation (S1 Table).

## Discussion

The present study demonstrated that thermal discharge from a NPP supported a specific fish community that included a higher number of tropical fish species. Although the difference in the mean temperature between the Nahagama and Otomi sites was only 1.3°C during the operation of the NPP, the fish community in the latter was distinctive. This may be due to tropical fishes generally having a lethal low temperature of 10–13°C, and therefore the small increase in temperature during winter conditions would support the survival of tropical vagrants [27, 28]. This study thus implies that a small increase in water temperature over the next few decades in temperate Japan will likely induce a drastic change in fish and other faunal communities from temperate to tropical species, together with a loss of algal vegetation.

In the Sezaki site, where a CPP was in operation, the sea water temperature was equivalent to the control site (Nagahama) and, therefore, did not support a community of tropical fishes. CPPs generally have a higher energy efficiency [29], and the temperature of the thermal discharge is lower compared to NPPs.

Geographic differences among the three sites should also be considered as a potential factor affecting fish communities in addition to the presence of a NPP. The Nagahama site is located in a highly protected bay, the Sezaki site is located outside the same bay and is exposed to waves, and the Otomi site is near the mouth of another bay (Fig 1). Common habitat characteristics of the relatively exposed locations of the Sezaki and Otomi sites may have contributed to the closer characteristics of fish assemblages represented in the nMDS (Fig 5). The relatively shallow depth of the bay where the Otomi site was located resulted in a markedly low temperature after the suspension of the Takahama NPP. The Otomi site was consistently shallow,

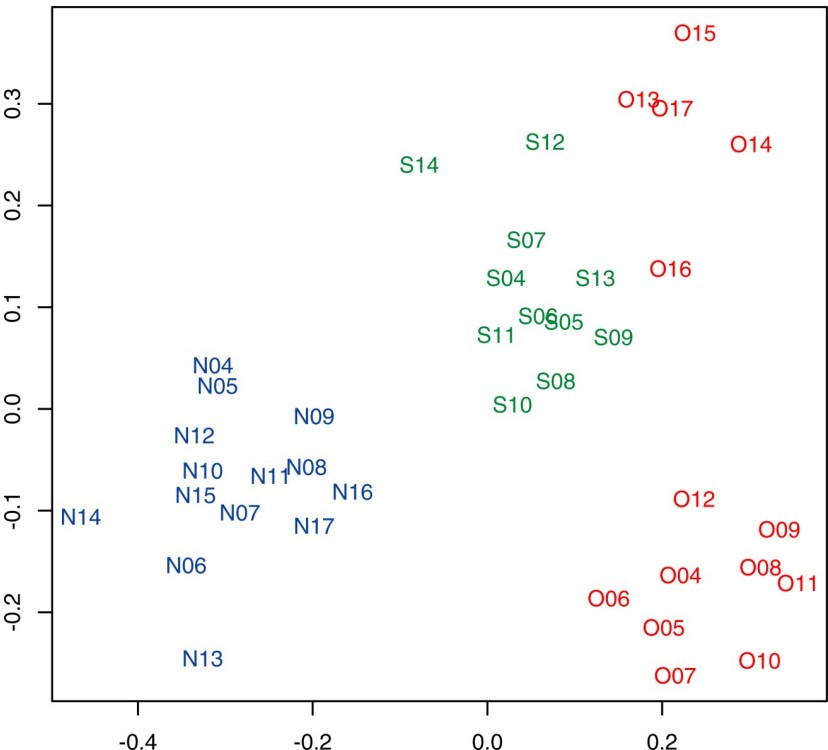

**Fig 5. Non-metric multi-dimensional scaling plot depicting fish assemblages in each site and year.** Each point represents the data for a surveyed site in a given year, e.g., N13 represents the data for the Nagahama site in 2013. Different colors and letters correspond to the different surveyed sites: O (red), Otomi; S (green), Sezaki; and N (blue), Nagahama.

whereas the other two sites had more variable depths in the survey area; such differences also have potentially confounding effects on comparisons. Future studies should include sites with different NPPs that may counterbalance such geographical differences.

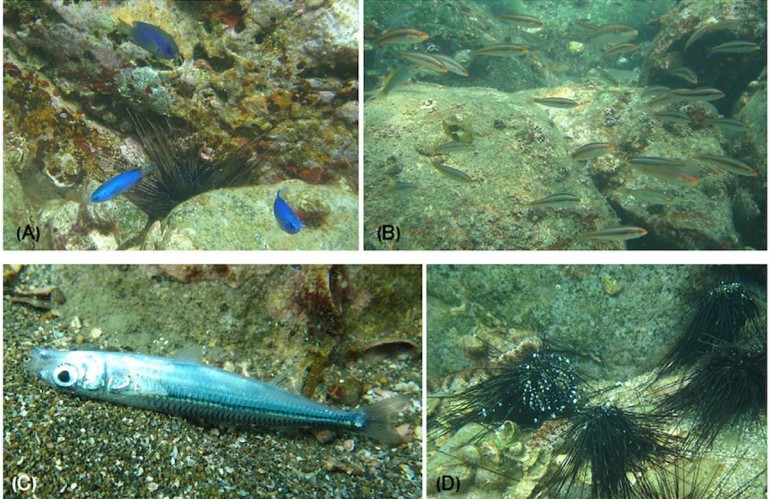

**Fig 6.** *P. coelestis* (A) and *S. interrupta* (B), both tropical fish species commonly found in the Otomi site when the Takahama nuclear power plant was in operation. Die off of a *H. tsurugae* (C) and *Diadema* sp. (D) after the suspension of the NPP. All the pictures were taken by the author.

Although most of the tropical fishes found in the Otomi site were likely to have been dispersed at the larval stage from the southern areas via the Tsushima Current, some of them may have been offspring of settled parental fish in this area. Both *P. coelestis* and *S. interrupta* were consistently found in the Otomi site. The minimum size of sexual maturation for *P. coelestis* and *S. interrupta* are reported to be 3.6 cm [30] and 6.1 cm [31], respectively. A substantial number of individuals of these species were above these sizes and thus were large enough to be considered mature. The northernmost population of *P. coelestis* was reported in Kominato, Chiba Prefecture (35˚07'N, 140˚11'E), where water temperature drops to 12˚C in winter [32]. Thus, overwintering would only be possible with the help of thermal discharge in this area. Locally spawned larval fish of these species may have succeeded in recruiting to this local warming site. The possibility of such self-recruitment can be tested using a molecular genetic approach such as the application of microsatellite markers [33].

Under the influence of thermal discharge, the fish community was found to have a high species richness including tropical species, and also a distinct range of species from those found in the lower latitude areas along the Japanese Archipelago. For instance, the author of the present study also conducted a visual census in the Tsushima Islands (34˚05–29'N, 129˚13–21'E) in the Sea of Japan and found that the community included large benthic predatory fishes such as lionfish *Pterois lunulata* and kelp groupers *Epinephelus bruneus* [34]. The lack of coral reef, vegetation, or other vertical structures in the Otomi site may have hindered the settlement of large predators. The absence of large predators is expected to have accelerated the loss of the seaweed bed, as grazers tend to suspend feeding when under the threat of potential predators [35]. Therefore, even with a warming trend, it is of paramount importance to focus attention on the maintenance of a healthy ecosystem that will include ecological engineers, such as corals and echinoderms, and top predators.

The present study was conducted by underwater visual censuses by a single observer which ensured consistency between locations and years. Visual censuses do, however, tend to have relatively low detectability for highly mobile or cryptic species [36]. Higher resolution censuses can be attained through environmental DNA analysis of filtered sea water, wherein fish species for both those recorded and missed by visual census can be detected [37].

Approximately a third of the electricity supply in Japan was dependent on nuclear power until the major accident in Fukushima Daiich NPP in 2011 [38], after which all the NPPs in Japan were suspended for inspection. In 2015 and thereafter, a few of the NPPs began operating again, including the Takahama NPP in May 2017. The possibility of a catastrophic disaster and the shortage of repository sites for radioactive waste are relatively well-recognized negative aspects of NPPs [39]. The pelagic larval stages of fishes such as European eel *Anguilla anguilla* and herring *Clupea harengus*, as well as other zooplankton can be impinged and killed in the cooling system of NPPs [40]. Thermal discharge may be environmentally destructive especially if many NPPs are in operation in a limited area, such as in the Sea of Japan (Fig 1). This area is known for much faster ocean warming (1.70˚C per century) compared to the global mean (0.53˚C per century). Although this may be partly due to the effect of the warm Tsushima Current, as is reported in other hot spots in south-eastern Australia [41], thermal discharge from many NPPs may spur the warming trend in some part of the Sea of Japan. Decision makers should, therefore, consider the ecological impacts of NPPs and base decisions regarding the operation of NPPs on long-term perspectives.

## Supporting information

**S1 Table. Abundance of each fish species encountered in Otomi, Sezaki, and Nagahama in each census.** Southern and northern limit of distribution (SLD and NLD, based on Nakabo

[19]) and their mean [= COD (center of distribution in the northern hemisphere)] are shown for each species. Each species was categorized into Temp (= temperate), Subtrop (= subtropical), or Trop (= tropical) according to the FishBase [24]. The codes each represent a census site and year; e.g., O04 represents Otomi in the year 2004. Bottom water temperature (Temp.) was recorded during each census.

(XLS)

## Acknowledgments

The underwater censuses were conducted with the help of many colleagues, particularly Kenji Minami, Kohji Takahashi, Hideki Sawada, and Mizuki Ogata. I also thank Dr. Yutaka Osada (National Research Institute of Fisheries Science) for statistical advice. Dr. Yoshiaki Kai (Kyoto University) provided constructive comments on the earlier version of the manuscript. Comments from Dr. Heather Patterson and anonymous reviewers substantially improved the quality of the manuscript.

## Author Contributions

**Conceptualization:** Reiji Masuda.

**Data curation:** Reiji Masuda.

**Formal analysis:** Reiji Masuda.

**Funding acquisition:** Reiji Masuda.

**Investigation:** Reiji Masuda.

**Methodology:** Reiji Masuda.

**Project administration:** Reiji Masuda.

**Resources:** Reiji Masuda.

**Validation:** Reiji Masuda.

**Visualization:** Reiji Masuda.

**Writing – original draft:** Reiji Masuda.

**Writing – review & editing:** Reiji Masuda.

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
