## [Decision Letter · Decision Letter 0]

28 Nov 2019

PONE-D-19-30294

Tropical fishes vanished after the operation of a nuclear power plant was suspended in the Sea of Japan

PLOS ONE

Dear Dr. Masuda,

Thank you for submitting your manuscript to PLOS ONE. After careful consideration, we feel that it has merit but does not fully meet PLOS ONE’s publication criteria as it currently stands. Therefore, we invite you to submit a revised version of the manuscript that addresses the points raised during the review process.

While the statistical rigor of the manuscript has been improved, both reviewers have identified issues with the paper, mainly around the statistical analyses, that need to be rectified. Reviewer 1 has noted that the PERMANOVA design should be the same as the univariate design and also has provided a number of comments to clarify the text, which is often confusing. Reviewer 2, who has not seen the manuscript before, has identified more serious issues with the statistics, including that some of the analyses are unbalanced due to missing data and that it appears the author did not follow the advice of the statistical advisor they consulted and tests for differences between time periods (i.e. when Otomi was operating and when it wasn't) were not done. In addition, the reviewer has noted that where the data are unbalance mixed model analyses are preferable to fixed effect ANOVAs, although these are more complicated.

I appreciate the effort the author has put into revising this manuscript and it certainly has improved. However, until concerns about the analyses are rectified the manuscript cannot be accepted. The author does not have to take the advice of the reviewer and present mixed model analyses, although that would improve the manuscript, but they must ensure that they are testing what needs to be tested and that they are not doing tests in an arbitrary fashion to validate visual differences, which what appears to have been done for the PERMANOVA.

I do want to see this work published so I would suggest the author again consult their statician to rectify the issues noted in the reviews. The author should also address the other comments of the reviewers around clarifying the text and presenting more detail about the methods, which is a bit confusing. 

We would appreciate receiving your revised manuscript by Jan 12 2020 11:59PM. To enhance the reproducibility of your results, we recommend that if applicable you deposit your laboratory protocols in protocols.io, where a protocol can be assigned its own identifier (DOI) such that it can be cited independently in the future. For instructions see: http://journals.plos.org/plosone/s/submission-guidelines#loc-laboratory-protocols

We look forward to receiving your revised manuscript.

Kind regards,

Heather M. Patterson, Ph.D.

Academic Editor

PLOS ONE

Journal Requirements:

1. Thank you for including your funding statement; "This study was partly supported by the CREST program from the Japan Science and Technology Agency (grant number: JPMJCR13A2; http://www.jst.go.jp/kisoken/crest/en/project/33/e33_13.html). The funders had no role in study design, data collection and analysis, decision to publish, or preparation of the manuscript. There was no additional funding received for this study."

Reviewers' comments:

Reviewer's Responses to Questions

**Comments to the Author**

1. Is the manuscript technically sound, and do the data support the conclusions?

Reviewer #1: Partly

Reviewer #2: Yes

2. Has the statistical analysis been performed appropriately and rigorously? 

Reviewer #1: No

Reviewer #2: No

3. Have the authors made all data underlying the findings in their manuscript fully available?

Reviewer #1: Yes

Reviewer #2: No

4. Is the manuscript presented in an intelligible fashion and written in standard English?

Reviewer #1: Yes

Reviewer #2: Yes

5. Review Comments to the Author

Reviewer #1: This manuscript describes changes in fish assemblages before and after the suspension of a nuclear power plant in the Sea of Japan. The article is generally well-written, however there are numerous issues regarding statistical analyses and interpretations. One specific issue that affects the overall concept of using the NPP as an indicator of future warming environment is that the temperature at NPP was only significantly different to reference sites in a single year prior to the suspension of NPP discharge (Fig 2a). I suspect this may relate to the lack of power from using only four temperature measurements at each site within each year, because the temps at NPP look considerably higher than the reference sites in Fig 2a. One way to avoid this would be to just present the temperature values, rather than statistically testing them, or pool temperature measurements before and after the suspension and make a single comparison for each site. I provide detailed comments below to assist a revision.

Title: replace ‘vanish’ with more scientifically-appropriate word

Line 33-42: Suggest moving this paragraph further down and opening the introduction with the information about climate change. The power plant is really just an opportunistic ‘method’ for investigating potential assemblage responses to increasing temperature sin the region.

Lines 51-54: Citations/references required to support these broad statements.

Line 75: suggest adding, “... as a proxy for understanding the impact of future temperature increases in the region”. Otherwise we are back to the purpose being about thermal discharge impacts, rather than what the authors state further up in the introduction regarding thermal plants simply acting as a model system for understanding climate change effects.

Lines 77-79: Figure 2a actually suggests there are few years where the temperature at the NPP site is significantly higher than the reference sites. In my opinion, the graph does depict a considerable difference in temperatures, and the standard errors are quite small, so I’m not sure why there are so few statistically-significant differences in temp. Perhaps values from different seasons

Line 119-120: what precisely was the decision based on – ease of sampling, depth, similar habitat etc?

Line 124: how many surveys were conducted at each site in each year?

Notes: Different depths – the NPP site is considerably shallower than either of the other two sites. Some discussion of this potentially confounding effect on comparisons between treatment and control sites is required.

Line 133-134: Rephrase this description, specifically “the criteria of relative abundance in the southern species”. Suggest the authors consider COD as an indicator of average latitudinal distribution within the assemblage, which when compared through time, can be used to understand trends in ‘tropicalisation’ of an assemblage.

Line 144-145: It is not possible to nest surveys within years – surveys represent the unit of replication, from what I can tell. The analysis does make sense if the authors nested years within periods, i.e. before and after suspension of the NPP.

Lines 142-145: More detail regarding the statistical design is required here. Were three separate ANOVAS conducted, one for each category? This seems to be the case form the wording of the results. So three separate three-factor analyses (one for each category), with factors of year (nested in period), period and site?

Line 149: what were the plotted data visually inspected for?

Lines 149-152: The PERMANOVA design should follow the same two-factor design used in the univariate comparisons. There is no (deductive) logic associated with plotting data, looking to see groupings (separations) and then constructing a statistical test to verify apparent visual differences, which is what the author appears to have done according to the description provided in this section.

Lines 155-156: Rephrase. The temperature actually appears to decrease linearly between 2009 and 2014. A substantial drop in temperature was also recorded during 2008, prior to the suspension of the NPP. Why?

Lines 163-165: I’m surprised by the low number of significant results, given the relatively small error indicated by the bars in Fig 2a. Perhaps Tukey’s HSD is too conservative here?

Line 164: significantly warmer than the reference sites I assume. Insert the clarification.

Lines 164-165: suggest rephrasing to, “…whereas no differences were found among sites in other years” for clarity.

Line 178-182: also need to state here that no significant differences were found between Otomi and the other reference site.

Lines 182-184: I recommend adding the results of the significant interaction term here in brackets, to support this statement regarding before/after. Interaction between sites and years is what you would expect if trends at one sites (hopefully NPP) differed to those at the other sites.

Lines 193: Which model term (factor) is this p-value for? See previous comments regarding more detail required on statistical design, but if species categories have been included as a factor in the analysis, I would expect to first see an interaction term reported, specifically the interaction between category and period. This would indicate a possible difference in trends of tropical species between the two time periods, but not subtropical and temperate species.

Lines 26-209: Again, no logical reason for structuring the analysis around the four groups seen in MDS. The statistical design should be decided a priori, similar to the univariate tests described for abundance and species richness.

Line 217-225: Suggest quantifying ‘dominance’ in some way, perhaps by reporting percentage of total fish abundance for each of these species.

Lines 229-234: worth mentioning here how many days after the NPP suspension the surveys were completed.

Reviewer #2: Reviewer Report for PONE-D-19-30294

Tropical fishes vanished after the operation of a nuclear power plant was suspended in the Sea of Japan PLOS ONE

Summary of Research:

The research examines a marine ecosystem that developed in the thermal outflow from a nuclear power station, and observed what happened when the artificial heat source was cut-off. These observations were compared with 2 other sites, one a coal fired power plant with smaller thermal output, and a control site with no artificial heat source.

Summary of impression:

My impression was of a unique study that entailed a great deal of detailed and well thought out work, with some strong findings. However, the analysis section is not optimal, and is incomplete. Presentation of findings should also be improved.

Major issues:

Statistical analyses are not optimal, to the point that some required treatment effects were not properly tested. Site*Year statistical models might be partly appropriate for Sezaki and Nagahama, but are not optimal for Otomi. The most appropriate models for Otomi need to test the 2 level time-period effects, which should also be applied to Sezaki and Nagahama to test if these sites differ in this respect to Otomi. I have more detailed suggestions below.

Minor issues:

Some detail is missing from the methods (P.7) that would otherwise clarify the areas that surveys were sampling at each site. Was each survey strip at each site selected at random, per survey instance, or was the one strip locations used at each site over all years? If transec strips are chosen at random at each site, what spatial area is being sampled to represent each site?.

The discussion could suggest bases for observed differences between Sezaki and Otomi, which might provide more perspective to the study. Otomi started in the 1970's and has high thermal output, Sezaki started in 2004 with smaller thermal output, Otomi had tropical fish species establish by ~30 years, and Sezaki didn't by 0-7 years.

Other comments:

Figures and Tables-

⦁ Figures would be improved by placing 95% confidence intervals around all means. Currently Fig 2 uses standard errors of means, and Fig 3 uses standard deviations, the former is better for purpose than the latter, but both are inferior to confidence intervals for allowing visual evaluation of results. One wouldn't need to conduct Tukey tests if 95% confidence intervals were used to highlight which treatment levels differed most, in situations where ANOVA's showed treatment levels differed significantly.

⦁ Tables. Table S2 reports 2-factor ANOVAs (site x year), which don't address all the needs of the study. The tests for differences between time periods within sites were not fitted and reported. By "time periods" I mean when Otomi was operating, and when it wasn't. The author did not adopt the advice of his own advisor, to fit these terms in his ANOVA models.

⦁ The plot of fish assemblages under multidimensional scaling (Fig 4) could be discussed in more detail. What variables were the X and Y axis dimensions mostly reflecting? The X axis dimension separates Nagahama from Sezaki and Otomi, and the Y axis doesn't. Related to this perhaps, is why may Otomi "off" attach to Sezaki, and is then furtherest from "Nagahama"?

Methods- Are generally well detailed.

Study Design- Was the best that could be done, for the study objectives. Being an observational study, it is difficult to fully control differences between sites.

Results and data- The results generally seemed to support the study conclusions. I didn't fully agree with the author, regarding his findings from Fig. 3 (P 10, lines 193-197), but I may have been misled by the SD bars in this figure.

I couldn't find any accessible temperature data, to accompany the observations on fish communities (latter provided).

References- References should be given for statistical tests, as well as an appropriate reference for the R-software.

Statistics-

Analysis of the data largely uses ANOVA, with output from the ANOVAs used to generate errors for means of interest, and to underlie Tukey's HSD tests. The ANOVA's that are reported are 2 factor Site*Year tests (Table S2). ANOVAs involving some form of nesting are briefly alluded to, but any findings from nested analyses were not reported (line 144, "surveys" under "site*years" were not analysed as nested effects). Year effects can be nested under time-periods, but these analyses were not conducted for the current manuscript. In these circumstances, reference to "nesting" should be removed from the manuscript.

The ANOVA's reported in the paper, don't optimally meet the needs of the study at Otomi, in that the factorial models site*year are of lesser use than testing time period effects (the years when NPP Otomi is operating and when it isn't). The site*year ANOVA is useful for generating SE's of means for within years between site effects (Fig. 2), but even for temperature, only 1 within year between site comparisons was found to be significant. The "time period" within site effects are better illustrated in Fig. 3, but ANOVA's to test these effects were presumably not conducted. It is unclear how the SD's for Fig. 3 were generated.

The experimental design for analyses with time period effects is not balanced for site, as Sezaki data is missing for years 2015-2017. As the Sezaki CPP wasn't operating in the winter of 2004, that years Sezaki data shouldn't be included in any time-period analysis either. Also, Otomi was switched off on Feb 20th 2012, in the middle of transec surveys, so subsequent 2012 data at this site should be dropped from the analysis, and depending how quickly a nuclear power plant can be shut down, and when this process started (if well before Feb 20th??), perhaps the Otomi 2012 data should be dropped entirely?. Of interest, was Otomi operating at the time of the winter surveys in 2008?, from Figure 2 it doesn't look as if it was. Where there is unbalanced data, mixed model analyses are much preferable to straight fixed effect ANOVA's.

Inspection of Fig. 2 suggests models in which to fit "time-period" effects for separate analyses for fish abundance, species richness, and temperature. Models might begin with a single "spline" across years and sites as a fixed effect, followed by "sites/time periods" as further fixed effects, and year as a random effect. Fitting fixed and random effects in a statistical model, complicates deterministic prediction of confidence intervals of means (for presentation in figures). Confidence intervals can then be derived stochastically, using the "sim" function in R, once a satisfactory mixed model is fitted. Benefits of such an approach are ability to test "time period" effects, have more powerful tests in the process than just "site*year", and to substitute 95% confidence intervals for presentation of results in figures, instead of using Tukey tests to identify which treatment levels are differing.

The models fitting time-period effects need to be analysed by an experienced statistician.

6. PLOS authors have the option to publish the peer review history of their article (what does this mean?). If published, this will include your full peer review and any attached files.

Reviewer #1: No

Reviewer #2: No

---

## [Author Response · Author response to Decision Letter 0]

13 Jan 2020

Academic Editor:

While the statistical rigor of the manuscript has been improved, both reviewers have identified issues with the paper, mainly around the statistical analyses, that need to be rectified. Reviewer 1 has noted that the PERMANOVA design should be the same as the univariate design and also has provided a number of comments to clarify the text, which is often confusing. Reviewer 2, who has not seen the manuscript before, has identified more serious issues with the statistics, including that some of the analyses are unbalanced due to missing data and that it appears the author did not follow the advice of the statistical advisor they consulted and tests for differences between time periods (i.e. when Otomi was operating and when it wasn't) were not done. In addition, the reviewer has noted that where the data are unbalance mixed model analyses are preferable to fixed effect ANOVAs, although these are more complicated.

Response: Thank you for identifying the flaws in statistics. PERMANOVA has been conducted in the form of pre- and post-suspension comparison for each site as suggested by Reviewer 1. Generalized Linear Mixed Model (GLMM) has been conducted in the revised manuscript with a help of a statistician. 

Reviewer #1: General Comments:

This manuscript describes changes in fish assemblages before and after the suspension of a nuclear power plant in the Sea of Japan. The article is generally well-written, however there are numerous issues regarding statistical analyses and interpretations. One specific issue that affects the overall concept of using the NPP as an indicator of future warming environment is that the temperature at NPP was only significantly different to reference sites in a single year prior to the suspension of NPP discharge (Fig 2a). I suspect this may relate to the lack of power from using only four temperature measurements at each site within each year, because the temps at NPP look considerably higher than the reference sites in Fig 2a. One way to avoid this would be to just present the temperature values, rather than statistically testing them, or pool temperature measurements before and after the suspension and make a single comparison for each site. I provide detailed comments below to assist a revision.

Response: 

I thank the reviewer for this general comment. At the first sentence of the result, I presented the temperature values to show the drop of temperature with the suspension of NPP. GLMM analysis has also supported the abrupt temperature change in 2012. I found an error in applying Tukey’s test in my previous analyses. Since GLMM is suitable for the analysis for the general result presented in Fig 2, ANOVA for these data were deleted.

Specific comments: 

Title: replace ‘vanish’ with more scientifically-appropriate word

Response: 

I consider this suggestion very carefully. I could replace “vanished” with “disappeared”. However, I think that the abrupt disappearance of tropical species may be expressed better with the term “vanish”. 

Line 33-42: Suggest moving this paragraph further down and opening the introduction with the information about climate change. The power plant is really just an opportunistic ‘method’ for investigating potential assemblage responses to increasing temperature sin the region.

Response:

Introduction has been changed to open with the general notion of global warming as is suggested by the reviewer.

Lines 51-54: Citations/references required to support these broad statements.

Response:

Following references have been cited as suggested:

1. Sortes CJB, Williams SL, Carlton JT. Marine range shifts and species introductions: comparative spread rates and community impact. Global Ecol Biogeogr 2010; 19: 303-316.

2. Brander KM. Global fish production and climate change. P Natl Acad Sci 2007; 104: 19709–19714.

Line 75: suggest adding, “... as a proxy for understanding the impact of future temperature increases in the region”. Otherwise we are back to the purpose being about thermal discharge impacts, rather than what the authors state further up in the introduction regarding thermal plants simply acting as a model system for understanding climate change effects.

Response:

Revision has been made according to the suggestion. This way general introduction starting with climate change is tightly connected with the study on thermal discharge impacts.

Lines 77-79: Figure 2a actually suggests there are few years where the temperature at the NPP site is significantly higher than the reference sites. In my opinion, the graph does depict a considerable difference in temperatures, and the standard errors are quite small, so I’m not sure why there are so few statistically-significant differences in temp. Perhaps values from different seasons

Response:

I found some errors in my previous Tukey’s test analyses. In the revised manuscript the focus is on the GLMM here. This way the temperature difference has become highly significant between pre- and post- suspension periods.

Line 119-120: what precisely was the decision based on – ease of sampling, depth, similar habitat etc?

Response:

The area was decided to maximize the detection of rare species in limited diving time. This has been clarified in the revised manuscript (L. 123-125) as follows:

The size of survey areas was decided to maximize detection of rare species within limited diving time based on previous studies conducted at the Nagahama and Otomi sites for other purposes [4, 18].

Line 124: how many surveys were conducted at each site in each year?

Response:

I conducted four surveys at each site in each year. This has been described further up in this section (L. 104).

Notes: Different depths – the NPP site is considerably shallower than either of the other two sites. Some discussion of this potentially confounding effect on comparisons between treatment and control sites is required.

Response:

Average depths were comparable between the Otomi and Nagahama site, but the depth of the latter was more variable. The potential confounding effect of such differences has been mentioned in the revised manuscript (L. 302-304) as follows:

The Otomi site was consistently shallow, whereas other two sites had more variable depths in the survey area; such differences also have potentially confounding effect on comparisons.

Line 133-134: Rephrase this description, specifically “the criteria of relative abundance in the southern species”. Suggest the authors consider COD as an indicator of average latitudinal distribution within the assemblage, which when compared through time, can be used to understand trends in ‘tropicalisation’ of an assemblage.

Response:

I thank the reviewer for notifying this ambiguity and redundancy. The logic in the previous description may have been circulatory. The description has been revised as “the criteria of spatiotemporal tropicalization” (L. 138-139).

Line 144-145: It is not possible to nest surveys within years – surveys represent the unit of replication, from what I can tell. The analysis does make sense if the authors nested years within periods, i.e. before and after suspension of the NPP.

Response:

I thank the reviewer for identifying this misuse of terms. The wording has been revised as suggested as follows (L. 152-153):

(i.e., survey years were nested in each period).

Lines 142-145: More detail regarding the statistical design is required here. Were three separate ANOVAS conducted, one for each category? This seems to be the case form the wording of the results. So three separate three-factor analyses (one for each category), with factors of year (nested in period), period and site?

Response:

I had conducted four separate two factor ANOVAs with factors of years and sites for the analysis in Fig. 2. I found that Tukey’s test was improper for this analysis. In the revised manuscript I conducted analyses with generalized linear mixed model (GLMM) to confirm that all response variables changed between the two periods (up to 2011 in temperature, species richness and center of distribution and up to 2012 in fish abundance). Then comparisons have been focused on the comparison between pre- and post-suspension period in the following sections. The statistical design has been detailed in the revised manuscript.

Line 149: what were the plotted data visually inspected for?

Response:

The above description has been deleted in the revised manuscript.

Lines 149-152: The PERMANOVA design should follow the same two-factor design used in the univariate comparisons. There is no (deductive) logic associated with plotting data, looking to see groupings (separations) and then constructing a statistical test to verify apparent visual differences, which is what the author appears to have done according to the description provided in this section.

Response:

The PERMANOVA design has been revised to compare two periods in each site according to the suggestion (L. 157-158).

Lines 155-156: Rephrase. The temperature actually appears to decrease linearly between 2009 and 2014. A substantial drop in temperature was also recorded during 2008, prior to the suspension of the NPP. Why?

Response:

The description of the temperature decrease has been revised to be based on facts referring raw data (L. 163-165). Relatively low temperature in 2008 may be due to heavy snow fall in this year. 

Lines 163-165: I’m surprised by the low number of significant results, given the relatively small error indicated by the bars in Fig 2a. Perhaps Tukey’s HSD is too conservative here?

Response:

I found a misuse of Tukey’s test here, so that the suggested part has been deleted in the revised manuscript.

Line 164: significantly warmer than the reference sites I assume. Insert the clarification.

Response:

The description has been removed in the revised manuscript.

Lines 164-165: suggest rephrasing to, “…whereas no differences were found among sites in other years” for clarity.

Response:

This phrase has also been removed.

Line 178-182: also need to state here that no significant differences were found between Otomi and the other reference site.

Response:

This phrase has also been removed.

Lines 182-184: I recommend adding the results of the significant interaction term here in brackets, to support this statement regarding before/after. Interaction between sites and years is what you would expect if trends at one sites (hopefully NPP) differed to those at the other sites.

Response:

ANOVA for this part has been deleted.

Lines 193: Which model term (factor) is this p-value for? See previous comments regarding more detail required on statistical design, but if species categories have been included as a factor in the analysis, I would expect to first see an interaction term reported, specifically the interaction between category and period. This would indicate a possible difference in trends of tropical species between the two time periods, but not subtropical and temperate species.

Response:

In this section ANOVA was applied for each category of fishes, i.e., tropical, subtropical or temperate fishes, in each site separately. Therefore, p-value was only for the comparison between pre- and post- suspension periods of NPP. This has been clarified in the method and result descriptions (L. 149-153 and 213-220).

Lines 26-209: Again, no logical reason for structuring the analysis around the four groups seen in MDS. The statistical design should be decided a priori, similar to the univariate tests described for abundance and species richness.

Response:

This section has been revised so that comparisons were conducted between two periods in each site.

Line 217-225: Suggest quantifying ‘dominance’ in some way, perhaps by reporting percentage of total fish abundance for each of these species.

Response:

Percentages of total abundance in these dominant species have been reported in the revised manuscript as suggested by the reviewer (L. 243-254).

Lines 229-234: worth mentioning here how many days after the NPP suspension the surveys were completed

Response:

Tropical species were not found in any winter for the following 5 years after the survey. Then the NPP restarted operation in May 2017. I am still continuing the survey and found that tropical species are gradually settling. This will be another fish to fry after the publication of the present paper.

Reviewer #2: Reviewer Report for PONE-D-19-30294

Tropical fishes vanished after the operation of a nuclear power plant was suspended in the Sea of Japan PLOS ONE

Summary of Research:

The research examines a marine ecosystem that developed in the thermal outflow from a nuclear power station, and observed what happened when the artificial heat source was cut-off. These observations were compared with 2 other sites, one a coal fired power plant with smaller thermal output, and a control site with no artificial heat source.

Summary of impression:

My impression was of a unique study that entailed a great deal of detailed and well thought out work, with some strong findings. However, the analysis section is not optimal, and is incomplete. Presentation of findings should also be improved.

Response:

I thank the reviewer for evaluating this work. The analyses have been substantially improved in the revised manuscript according to the suggestions.

Major issues:

Statistical analyses are not optimal, to the point that some required treatment effects were not properly tested. Site*Year statistical models might be partly appropriate for Sezaki and Nagahama, but are not optimal for Otomi. The most appropriate models for Otomi need to test the 2 level time-period effects, which should also be applied to Sezaki and Nagahama to test if these sites differ in this respect to Otomi. I have more detailed suggestions below.

Response:

I agree with the reviewer that Site*Year model was not optimal for the goal of this study. Thus, the above analysis was abandoned and GLMM has been applied in the revised manuscript.

Minor issues:

Some detail is missing from the methods (P.7) that would otherwise clarify the areas that surveys were sampling at each site. Was each survey strip at each site selected at random, per survey instance, or was the one strip locations used at each site over all years? If transec strips are chosen at random at each site, what spatial area is being sampled to represent each site?.

The discussion could suggest bases for observed differences between Sezaki and Otomi, which might provide more perspective to the study. Otomi started in the 1970's and has high thermal output, Sezaki started in 2004 with smaller thermal output, Otomi had tropical fish species establish by ~30 years, and Sezaki didn't by 0-7 years.

Response:

Methods of the survey has been described in more detail in the revised manuscript (L 119-125). The survey strip is not totally random, yet not totally fixed. This way the detection of relatively rare species was maximized in limited diving time.

For the detailed comparison among survey sites, I think there are some limitations. No data was available concerning what occurred in Otomi during 1970’s, and Sezaki differed from Otomi both in the thermal output and geographic location. Therefore, in this manuscript, I would like to refrain deeper discussions to compare the sites other than the suspension of NPP.

Other comments:

Figures and Tables-

⦁ Figures would be improved by placing 95% confidence intervals around all means. Currently Fig 2 uses standard errors of means, and Fig 3 uses standard deviations, the former is better for purpose than the latter, but both are inferior to confidence intervals for allowing visual evaluation of results. One wouldn't need to conduct Tukey tests if 95% confidence intervals were used to highlight which treatment levels differed most, in situations where ANOVA's showed treatment levels differed significantly.

Response:

Error bars in the previous Fig 3 have been replaced by standard errors as suggested. I was not sure how to draw 95% confidence limit with the graphic software I use (Kaleida Graph).

⦁ Tables. Table S2 reports 2-factor ANOVAs (site x year), which don't address all the needs of the study. The tests for differences between time periods within sites were not fitted and reported. By "time periods" I mean when Otomi was operating, and when it wasn't. The author did not adopt the advice of his own advisor, to fit these terms in his ANOVA models.

Response:

ANOVA corresponding to Fig 2 was not efficient. Therefore, GLMM has been applied here to confirm the difference between two periods. Time period effect on each variable as well as interaction have been reported based on the GLMM results in the new Table 1 in the revised manuscript. Then the fish assemblages in two periods were further compared with the new Fig 4. I hope this makes more sense.

⦁ The plot of fish assemblages under multidimensional scaling (Fig 4) could be discussed in more detail. What variables were the X and Y axis dimensions mostly reflecting? The X axis dimension separates Nagahama from Sezaki and Otomi, and the Y axis doesn't. Related to this perhaps, is why may Otomi "off" attach to Sezaki, and is then furtherest from "Nagahama"?

Response:

I thank the reviewer for notifying these interesting aspects. In my understanding it is not possible to detect what factor is contributing to each axis in nMDS. The reason Otomi was off attached to Sezaki and far from Nagahama in the nMDS may be that Nagahama is located in highly protected inner bay, while location of other two sites are relatively exposed, as is seen in Fig 1. This is included in the revised discussion as follows (L 298-300):

Common character of relatively exposed location in the Sezaki and the Otomi sites may have contributed to the closer characteristics of fish assemblages represented in the nMDS (Fig 5).

Methods- Are generally well detailed.

Study Design- Was the best that could be done, for the study objectives. Being an observational study, it is difficult to fully control differences between sites.

Results and data- The results generally seemed to support the study conclusions. I didn't fully agree with the author, regarding his findings from Fig. 3 (P 10, lines 193-197), but I may have been misled by the SD bars in this figure.

Response:

SD bars have been replaced by SE bars. There were some flaws in the plotting of SD in the previous version of the manuscript. They are all rectified. SE bars are now consistent with p-values in the revised new Fig 4.

I couldn't find any accessible temperature data, to accompany the observations on fish communities (latter provided).

Response:

Temperature data of all the survey sites and dates have been included in the 5th line of the S1 Table.

References- References should be given for statistical tests, as well as an appropriate reference for the R-software.

Response:

Five references are given for R-software itself and install packages in the revised manuscript (References 21-25).

Statistics-

Analysis of the data largely uses ANOVA, with output from the ANOVAs used to generate errors for means of interest, and to underlie Tukey's HSD tests. The ANOVA's that are reported are 2 factor Site*Year tests (Table S2). ANOVAs involving some form of nesting are briefly alluded to, but any findings from nested analyses were not reported (line 144, "surveys" under "site*years" were not analysed as nested effects). Year effects can be nested under time-periods, but these analyses were not conducted for the current manuscript. In these circumstances, reference to "nesting" should be removed from the manuscript.

The ANOVA's reported in the paper, don't optimally meet the needs of the study at Otomi, in that the factorial models site*year are of lesser use than testing time period effects (the years when NPP Otomi is operating and when it isn't). The site*year ANOVA is useful for generating SE's of means for within years between site effects (Fig. 2), but even for temperature, only 1 within year between site comparisons was found to be significant. The "time period" within site effects are better illustrated in Fig. 3, but ANOVA's to test these effects were presumably not conducted. It is unclear how the SD's for Fig. 3 were generated.

Response:

I thank these comments by the reviewer. Tukey’s test in Fig 2 did not make much sense so I deleted them. I did conduct ANOVA for the analysis in the previous Fig 3, the results of which have been more detailed in the revised manuscript.

The experimental design for analyses with time period effects is not balanced for site, as Sezaki data is missing for years 2015-2017. As the Sezaki CPP wasn't operating in the winter of 2004, that years Sezaki data shouldn't be included in any time-period analysis either. Also, Otomi was switched off on Feb 20th 2012, in the middle of transec surveys, so subsequent 2012 data at this site should be dropped from the analysis, and depending how quickly a nuclear power plant can be shut down, and when this process started (if well before Feb 20th??), perhaps the Otomi 2012 data should be dropped entirely?. Of interest, was Otomi operating at the time of the winter surveys in 2008?, from Figure 2 it doesn't look as if it was. Where there is unbalanced data, mixed model analyses are much preferable to straight fixed effect ANOVA's.

Inspection of Fig. 2 suggests models in which to fit "time-period" effects for separate analyses for fish abundance, species richness, and temperature. Models might begin with a single "spline" across years and sites as a fixed effect, followed by "sites/time periods" as further fixed effects, and year as a random effect. Fitting fixed and random effects in a statistical model, complicates deterministic prediction of confidence intervals of means (for presentation in figures). Confidence intervals can then be derived stochastically, using the "sim" function in R, once a satisfactory mixed model is fitted. Benefits of such an approach are ability to test "time period" effects, have more powerful tests in the process than just "site*year", and to substitute 95% confidence intervals for presentation of results in figures, instead of using Tukey tests to identify which treatment levels are differing.

The models fitting time-period effects need to be analysed by an experienced statistician.

Response:

I thank the reviewer for providing the idea of analysis. I consulted to a statistician, and according to his advice, GLMMs have been conducted in the revised manuscript as is represented in new Fig 3 and Table 1. This way the data in the year 2012 have been made use of.

---

## [Decision Letter · Decision Letter 1]

18 Feb 2020

PONE-D-19-30294R1

Tropical fishes vanished after the operation of a nuclear power plant was suspended in the Sea of Japan

PLOS ONE

Dear Dr. Masuda,

Thank you for submitting your manuscript to PLOS ONE. After careful consideration, we feel that it has merit but does not fully meet PLOS ONE’s publication criteria as it currently stands. Therefore, we invite you to submit a revised version of the manuscript that addresses the points raised during the review process.

I appreciate the huge amount of work the author has done to improve this manuscript. And I think it certainly has improved. As noted by Reviewer 2, I think this decision is halfway between major and minor revision required. I think the text has greatly improved and the use of GLMMs in the paper has improved the analyses. I think the problem with the paper is that is it not entirely clear what exactly has been done and the presentation of the results could be better. So I think the data and the story are there, it is just making it very clear that is important. As currently written it is quite confusing.

Reviewer 2 in particular has provided very detailed comments and suggestions to improve the analyses and the explanation of what was done. They have also provided example code to improve the presentation of the results. I know this paper has been reviewed quite a few times already, but I think it is getting very close to where it needs to be. Again, the author has at least attempted to undertake all the comments and suggestions provided so that is appreciated. I encourage the author to consider all the comments provided when making their revisions.

We would appreciate receiving your revised manuscript by Apr 03 2020 11:59PM. To enhance the reproducibility of your results, we recommend that if applicable you deposit your laboratory protocols in protocols.io, where a protocol can be assigned its own identifier (DOI) such that it can be cited independently in the future. For instructions see: http://journals.plos.org/plosone/s/submission-guidelines#loc-laboratory-protocols

We look forward to receiving your revised manuscript.

Kind regards,

Heather M. Patterson, Ph.D.

Academic Editor

PLOS ONE

Reviewers' comments:

Reviewer's Responses to Questions

**Comments to the Author**

1. If the authors have adequately addressed your comments raised in a previous round of review and you feel that this manuscript is now acceptable for publication, you may indicate that here to bypass the “Comments to the Author” section, enter your conflict of interest statement in the “Confidential to Editor” section, and submit your "Accept" recommendation.

Reviewer #1: (No Response)

Reviewer #2: (No Response)

2. Is the manuscript technically sound, and do the data support the conclusions?

Reviewer #1: Partly

Reviewer #2: Yes

3. Has the statistical analysis been performed appropriately and rigorously? 

Reviewer #1: I Don't Know

Reviewer #2: No

4. Have the authors made all data underlying the findings in their manuscript fully available?

Reviewer #1: Yes

Reviewer #2: Yes

5. Is the manuscript presented in an intelligible fashion and written in standard English?

Reviewer #1: Yes

Reviewer #2: Yes

6. Review Comments to the Author

Reviewer #1: Line numbers referenced here refer to the track-changed version.

I’ve reviewed this work multiple times. Although I think the approach is gradually progressing toward a correct statistical design, the description of the GLMM methods and results are still unclear, particularly with respect the temporal component. My best guess is that the author has used a model term of “period” (although I think they call this “Year2”, which is confusing), which has two levels. Then they have created many models, each of which splits the dataset at a different calendar year. So, in one model, “period” might have one level with 2004-2008 and the other with 2009 onwards, while in another model, one level might span 2004-2012 and the other level 2013 onwards. AICs are then used to compare all these temporal splits, along with a model with no split. I think this is what the author refers to as the “bisection method”, and seems to correspond to Figure 3, where you can see an AIC value provided for each calendar year. Although my interpretation here is somewhat contradicted by this statement - “(Variable ~ Site + Year2 + Site:Year2 + (1|Survey), REML = FALSE) where Year2 = 1*(Year > 237 2012) for fish abundance and Year2 = 1*(Year > 2011) for other variables” (Lines 236-237), which implies that the time variable has a set calendar cut-off depending on the response variable (fish abundance or other). But then how is there AIC value for each calendar year (Figure 3)? Hence my confusion.

If my above assessment is correct, it raises some concerns regarding the balance of the design, and the testing of other model terms. Regarding balance, those models that have only one or two years’ data in one level, and many in the other, are likely to have issues with power. Has this been considered? Regarding testing of other model terms, why only use AIC for the models that differ in the “Year2” model term and not those of most interest, i.e. the interaction between site and year? As far as I can tell, the importance of these model terms is assessed using p-values alone. A preferable approach would be to use AIC throughout, and simply split the “Year2” variable into “before” and “after”, based on the known year the NPP stopped operation. This would provide only two models to compare for this term, with and without “Year2”. These two models can then be compared against models that vary the inclusion of other terms, e.g. with and without the term “Site:Year2”, which is what you are really interested in. If my above assessment of the modelling approach is incorrect, the methodological description requires considerable improvement to clarify exactly what was done, how many models were used, the years they covered etc.

For the analyses of species categories (tropical, temperate etc), why revert to nested ANOVAs within each site? The same GLMM approach used for total abundance, where site is a factor with three levels and an interaction term between site and year is included, seems warranted for each of these sub-categories, unless a meaningful combination response variable could be derived (e.g. “tropical proportional abundance”), which would reduce the number of separate analyses required.

More detail regarding the PERMANOVA modelling is required. Was a random nested factor “year” included in the modelling? How were individual surveys within years dealt with?

Reviewer #2: Referee Report

The author has adopted many of the suggestions from the preceeding reviews of his manuscript.

My recommendation is more midway between major and minor revision, than simply major revision.

The author has improved analysis, but not quite complete yet. Given the options, I had to give a "no".

General

More description has been provided of sample data collection from survey sites.

Questions on the multidimensional scaling analysis that is presented in Fig. 5, have been answered (but not for the manuscript).

Temperature data for the 3 marine sites has been provided in Table S1. This was very helpful for me to try mixed model analysis and presentation of findings (see attachment).

References have been provided for some of the statistical tests described in the methods section. However, could references for the multi-dimensional scaling analysis be provided?, and for the Bisection analysis? (if retained)

On line 193, should "year" be "year2"?. Should "year2" be better named throughout, to indicate time period for Otomi NPP state of operation ("on"/"off")?.

Statistical analysis

The manuscript now seeks to directly tackle analysis of differences in response variables across the time periods when the Otomi NPP was operating and when it was not. That is, 2-factor ANOVA's were dropped, and mixed model analyses adopted instead. However, I believe further improvement to these analyses may be needed, if I correctly interpret what was done.

A thing to note is that the design for the main treatment effect -Otomi NPP operation- is unreplicated over the study. This is necessarily the case of course, given how the data could be collected, but it has implications to the analysis, because trend in response variables over years might be confounded with Otomi NPP operating period effects. The major implication is that the Otomi "on"/"off" effect can't be directly tested via a single statistical model. A way of get around this limitation is to use confidence intervals to depict differences between operating periods, and to show how sites differ when - Otomi NPP - is operating, and when it isn't. (Example figure to illustrate this is provided for temperature data; in attachment). [A useful reference to using 95% confidence intervals instead of p-values is Cumming G. (2012). Understanding The New Statistics. Effect Sizes, Confidence Intervals, and Meta Analysis. Routledge, Taylor & Francis Group.]

The mixed model used to analyse the study's data across the (Otomi NPP) operating periods, is given line 194. This is more R-code than a direct model description, and the actual variables used are not clear, eg. what is "Survey". "Survey" is mentioned previously on line 143, but is then not clearly enough defined for a description of the statistical model. I suggest a clearer description of the model, perhaps:-

response ~ Site (fixed Effect) + Year/survey (random Effects)

Separate analyses for NPP "on", and "off". "Survey" is the replicate data nested below Years; both as random effects in the model.

I think it is reasonable to separately analyse the data for when Otomi NPP is "on", and when it is "off", given this treatment is unreplicated. This has the advantage of simpler statistical models and analyses, and avoids any interaction complications when extracting parameter estimates from analyses. Any "Interaction" effects are then simply embedded in estimated means and confidence intervals, and can be evaluated visually while comparing sites for each response variable.

Tests of difference between fixed effect levels in mixed models are produced by the lme4 software, but the F tests reported are controversial among mixed-model authorities, Chi-square tests being better regarded, and these in turn are less well regarded than direct comparison of estimated confidence intervals (example for temperature data shown below). Data+models will differ in degree of these problems, but it might be best to adopt the best accepted presentation method. The author's reference, Bates et al (2015) covers these issues.

The "mixed model" analyses used for data analysis were described in the manuscript as GLMM's, I believe they might actually be LMM's. GLMM's use non-normal error terms and non linear "link" functions between data and model, and are even more problematic to interpret than LMM's. LMM's can accomodate transformations of the response variable (ie. sqrt, log, etc), without the analyses becoming GLMM's. I believe the "G" letter should be dropped from GLMM throughout this manuscript.

Bisection analyses

Perhaps partly in response to a previously suggested uncertainty about whether the Otomi NPP was operating at its previous level when the 2012 surveys were conducted, the author has included a "Bisection" analysis to test when the the Otomi (only?) data can be empirically broken into 2 uniform time periods. I may be misinterpreting the details around this in the manuscript. If retained, this content may need to be better introduced and wht it shows more clearly described. A reference for this analysis may also be useful. I didn't see any discussion of the outcome of the "Bisection" analyses, so my question is, how did these analyses improve the study's analysis and major findings?. This material may even weaken the study, if it shows changes in response variables preceeded changes to Otomi NPP being switched off. This content comes back to the question in my previous review, was the Otomi NPP being wound down well before the 2012 Otomi survey?, such that an effect could be seen in the 2012 survey at Otomi (for temperature and fish assemblages), or were the 2011/2012 effects at Otomi also partly seen at other sites, and therefore due to a more general environmental effect?. This content then should be more clearly discussed if retained in the manuscript.

If the Bisection analyses don't improve clarity toward resolving this issue of Otomi possibly being wound down prior to the 2012 survey, and details on the Otomi NPP power output (prior to the 2012 survey times) can't be accessed, should the "Bisection" analyses be retained in the manuscript?. If not, the data could just be analysed "as is" (despite the 2012 Otomi on/off uncertainty), strong effects in the study will still be demonstrated.

Presentation

Table 1.

Contains parameter details for site and "year2" (~NPP operating time intervals?). Some of this parameter information is also in Figure 4, so is presented in both "figures and table" (usually not acceptable), and so should be in one or the other. Table 1 does additionally contain p-value information for F and t tests of significance between fixed effect levels (ie. with site and "year2"), but these are poorly regarded (outlined above), and presents parameter estimates in a way that can be confusing to people not familiar with this statistical model format. The F and t tests could be dropped, and Chi-square tests for simply differences between "sites" could be presented instead. Parameter values for each site are then in an updated Fig. 4 (suggestion below), instead of in Table 1. Tests for differences between Otomi time periods (on/off) are not needed in models, if analyses for Otomi "on" and Otomi "off" are conducted separately (and reported separately within Table 1, with site estimates presented in Fig 4 (suggestion)). This reporting for each variable in the current Table 1, could be extended to the Tropical, Sub-tropical and Temperate species subdivisions within the Abundance and Richness variables. Tests for these subdivisions currently appear obscurely as labels in Figure 4, and introduce some clutter into the figure.

Figure 2.

This Figure contains un-necessary information in the form of standard errors for each year*site data point. With the dropping of the 2-factor ANOVA approach to analysis, this information is no longer really needed, and would instead be provided more directly in an updated Figure 4 (see below). However, this figure can still provide useful information, in depicting how the raw data varies over sites. For the case of the temperature data, adding lines to adjacent year means, gives a more clearly digestable idea of how sites can be quite consistently ranked for mean temperature over the series of years that were studied, and showing strongly the effect of the heat source being cut at Otomi around 2012 (and perhaps the same shown for "fish abundance", etc...).

So perhaps confidence intervals can be dropped from Fig. 2, and lines added linking adjacent year means for each site. This altered figure is especially useful for temperature, as this is driving the biological effects in the study.

Figure 3.

discussed above, still needed?

Figure 4.

Error bars presented in Figure 4 of the previous manuscript, were changed from standard deviations to standard errors. This is an improvement but still not optimal. I have included in an attachment, some R-code on how to calculate 95% confidence intervals for fixed effects from mixed models, by "simulating" mixed model fits (from lmer(model) ), and then extracting 0.025 and 0.975 percentiles. To illustrate, this was done for the temperaure data now included in Table S1, for when Otomi NPP was operating, and when it wasn't. The confidence intervals have been plotted in an attached Figure, generated by R (code attached). I recommend using 95% CI's, because significant differences can be noted directly from these plots, eliminating the need to provide information currently in Table 1. The author might need to reconsider grouping separate plots in Fig. 4 around variables (for greatest test validity), rather than sites, as is currently the case. Temperature confidence intervals could be included in Figure 4, or a separate small similarly structured Figure.

A rule of thumb applies to presentations comparing treatment means via 95% confidence intervals. Where 2 intervals just fail to overlap, the treatment levels differ at approximately p=0.01. Confidence intervals will differ at p~0.05 if the overlap is about a third of one side of an an error bar (Cumming (2012) ).

Please note that Figure 4 should plot both upper and lower confidence intervals, to enable quick visual comparisons to be made between "sites"/"treatments". Bar plots can be used to plot estimated means, or points, as in the temperature example provided, but any bar shading should obviously be light enough not to obscure the lower bars.

The author plotted standard errors in his new Figure 4, because his prefered plotting software (Kaleida Graph) couldn't draw 95% CI's. If Kaleida Graph does not have an option to plot user supplied errors (where this can't be directly generated within the software), other plotting software to do this is available. R can be programmed to produce such plots (see attached document for code), though time consuming to implement.

Attachment provided to this report.

7. PLOS authors have the option to publish the peer review history of their article (what does this mean?). If published, this will include your full peer review and any attached files.

Reviewer #1: No

Reviewer #2: No

---

## [Author Response · Author response to Decision Letter 1]

30 Mar 2020

Responses to Editor and Reviewer Comments

Academic Editor:

I appreciate the huge amount of work the author has done to improve this manuscript. And I think it certainly has improved. As noted by Reviewer 2, I think this decision is halfway between major and minor revision required. I think the text has greatly improved and the use of GLMMs in the paper has improved the analyses. I think the problem with the paper is that is it not entirely clear what exactly has been done and the presentation of the results could be better. So I think the data and the story are there, it is just making it very clear that is important. As currently written it is quite confusing.

Reviewer 2 in particular has provided very detailed comments and suggestions to improve the analyses and the explanation of what was done. They have also provided example code to improve the presentation of the results. I know this paper has been reviewed quite a few times already, but I think it is getting very close to where it needs to be. Again, the author has at least attempted to undertake all the comments and suggestions provided so that is appreciated. I encourage the author to consider all the comments provided when making their revisions.

Response: Thank you for your help in improving my manuscript. I read comments by reviewers repeatedly, purchased the recommended reference by the Reviewer 2 (“Understanding the new statistics”) and read through it, and made revisions on data analyses and presentations. Specifically, the bisection method model and GLMMs in the previous manuscript were removed together with Fig 3 and Table 1, and more simple liner mixed models have been adopted. New Fig 3 and improved Fig 2 and 4 with confidential intervals have been prepared. I also thank the editor for the editorial comments. All the flaws have been rectified in the revised manuscript.

Reviewer #1: General Comments:

I’ve reviewed this work multiple times. Although I think the approach is gradually progressing toward a correct statistical design, the description of the GLMM methods and results are still unclear, particularly with respect the temporal component. My best guess is that the author has used a model term of “period” (although I think they call this “Year2”, which is confusing), which has two levels. Then they have created many models, each of which splits the dataset at a different calendar year. So, in one model, “period” might have one level with 2004-2008 and the other with 2009 onwards, while in another model, one level might span 2004-2012 and the other level 2013 onwards. AICs are then used to compare all these temporal splits, along with a model with no split. I think this is what the author refers to as the “bisection method”, and seems to correspond to Figure 3, where you can see an AIC value provided for each calendar year. Although my interpretation here is somewhat contradicted by this statement - “(Variable ~ Site + Year2 + Site:Year2 + (1|Survey), REML = FALSE) where Year2 = 1*(Year > 237 2012) for fish abundance and Year2 = 1*(Year > 2011) for other variables” (Lines 236-237), which implies that the time variable has a set calendar cut-off depending on the response variable (fish abundance or other). But then how is there AIC value for each calendar year (Figure 3)? Hence my confusion.

If my above assessment is correct, it raises some concerns regarding the balance of the design, and the testing of other model terms. Regarding balance, those models that have only one or two years’ data in one level, and many in the other, are likely to have issues with power. Has this been considered? Regarding testing of other model terms, why only use AIC for the models that differ in the “Year2” model term and not those of most interest, i.e. the interaction between site and year? As far as I can tell, the importance of these model terms is assessed using p-values alone. A preferable approach would be to use AIC throughout, and simply split the “Year2” variable into “before” and “after”, based on the known year the NPP stopped operation. This would provide only two models to compare for this term, with and without “Year2”. These two models can then be compared against models that vary the inclusion of other terms, e.g. with and without the term “Site:Year2”, which is what you are really interested in. If my above assessment of the modelling approach is incorrect, the methodological description requires considerable improvement to clarify exactly what was done, how many models were used, the years they covered etc.

Response: The analysis using the bisection method model has been removed in the revised manuscript. This was because i) the model has a problem of balance having “only one or two years’ data in one level” as was suggested by the reviewer 1, and also ii) suspension of NPP as a treatment effect is “unreplicated” as was pointed out by the reviewer 2 so that there is not much point to conduct this analysis. Therefore, in the revised manuscript linear mixed models have been adopted separately during NPP on and off periods according to the suggestion by the reviewer 2. Providing 95% confidence intervals has made it possible to compare inter-period difference within a site and inter-site difference within a period at a glance in the new Figs 3 and 4.

Reviewer #1: Specific Comments:

For the analyses of species categories (tropical, temperate etc), why revert to nested ANOVAs within each site? The same GLMM approach used for total abundance, where site is a factor with three levels and an interaction term between site and year is included, seems warranted for each of these sub-categories, unless a meaningful combination response variable could be derived (e.g. “tropical proportional abundance”), which would reduce the number of separate analyses required.

Response: The analyses of species categories have also been revised to adopt linear mixed models to be consistent.

More detail regarding the PERMANOVA modelling is required. Was a random nested factor “year” included in the modelling? How were individual surveys within years dealt with?

Response: Abundance data were summed for each species in each year, rather than modelling nested PERMANOVA. This has been clarified in the revised manuscript (L 156-157). 

Reviewer #2: General Comments:

General

More description has been provided of sample data collection from survey sites.

Questions on the multidimensional scaling analysis that is presented in Fig. 5, have been answered (but not for the manuscript).

Temperature data for the 3 marine sites has been provided in Table S1. This was very helpful for me to try mixed model analysis and presentation of findings (see attachment).

References have been provided for some of the statistical tests described in the methods section. However, could references for the multi-dimensional scaling analysis be provided?, and for the Bisection analysis? (if retained)

On line 193, should "year" be "year2"?. Should "year2" be better named throughout, to indicate time period for Otomi NPP state of operation ("on"/"off")?.

Response: I thank the reviewer for valuable comments. The description of the multidimensional scaling analysis has been described more in detail in the revised manuscript (L 156-158). The reference for the multi-dimensional scaling analysis (Anderson 2001) has been cited, while bisection analysis has been deleted in the revised manuscript. 

Reviewer #2: Specific Comments:

Statistical analysis

The manuscript now seeks to directly tackle analysis of differences in response variables across the time periods when the Otomi NPP was operating and when it was not. That is, 2-factor ANOVA's were dropped, and mixed model analyses adopted instead. However, I believe further improvement to these analyses may be needed, if I correctly interpret what was done.

A thing to note is that the design for the main treatment effect -Otomi NPP operation- is unreplicated over the study. This is necessarily the case of course, given how the data could be collected, but it has implications to the analysis, because trend in response variables over years might be confounded with Otomi NPP operating period effects. The major implication is that the Otomi "on"/"off" effect can't be directly tested via a single statistical model. A way of get around this limitation is to use confidence intervals to depict differences between operating periods, and to show how sites differ when - Otomi NPP - is operating, and when it isn't. (Example figure to illustrate this is provided for temperature data; in attachment). [A useful reference to using 95% confidence intervals instead of p-values is Cumming G. (2012). Understanding The New Statistics. Effect Sizes, Confidence Intervals, and Meta Analysis. Routledge, Taylor & Francis Group.]

Response: I thank the reviewer for introducing Cumming (2012). I purchased the book and found it to be fascinating. I was surprised to see that none of statistical textbooks published in Japan, at least those I have checked, is referring this work. The concept described in the book seems sensible and quite revolutionary, yet I still hesitate to totally abandon the “null hypothesis significant testing” attitudes. I however have understood the usefulness of providing confidence intervals. Standard errors have been thus replaced by 95% confidence intervals in the revised manuscript. This way complications of p-value descriptions have been avoided in the revised manuscript. This method has been described in detail in the revised manuscript (L140-154).

The mixed model used to analyse the study's data across the (Otomi NPP) operating periods, is given line 194. This is more R-code than a direct model description, and the actual variables used are not clear, eg. what is "Survey". "Survey" is mentioned previously on line 143, but is then not clearly enough defined for a description of the statistical model. I suggest a clearer description of the model, perhaps:-

response ~ Site (fixed Effect) + Year/survey (random Effects)

Separate analyses for NPP "on", and "off". "Survey" is the replicate data nested below Years; both as random effects in the model.

I think it is reasonable to separately analyse the data for when Otomi NPP is "on", and when it is "off", given this treatment is unreplicated. This has the advantage of simpler statistical models and analyses, and avoids any interaction complications when extracting parameter estimates from analyses. Any "Interaction" effects are then simply embedded in estimated means and confidence intervals, and can be evaluated visually while comparing sites for each response variable.

Tests of difference between fixed effect levels in mixed models are produced by the lme4 software, but the F tests reported are controversial among mixed-model authorities, Chi-square tests being better regarded, and these in turn are less well regarded than direct comparison of estimated confidence intervals (example for temperature data shown below). Data+models will differ in degree of these problems, but it might be best to adopt the best accepted presentation method. The author's reference, Bates et al (2015) covers these issues.

The "mixed model" analyses used for data analysis were described in the manuscript as GLMM's, I believe they might actually be LMM's. GLMM's use non-normal error terms and non linear "link" functions between data and model, and are even more problematic to interpret than LMM's. LMM's can accomodate transformations of the response variable (ie. sqrt, log, etc), without the analyses becoming GLMM's. I believe the "G" letter should be dropped from GLMM throughout this manuscript.

Response: I checked references carefully and understood that linear mixed model rather than GLMM is more suitable for the present data set. Also the data analyses have been conducted separately in the NPP on and off periods in the revised manuscript as was suggested by the reviewer 2.

Bisection analyses

Perhaps partly in response to a previously suggested uncertainty about whether the Otomi NPP was operating at its previous level when the 2012 surveys were conducted, the author has included a "Bisection" analysis to test when the the Otomi (only?) data can be empirically broken into 2 uniform time periods. I may be misinterpreting the details around this in the manuscript. If retained, this content may need to be better introduced and wht it shows more clearly described. A reference for this analysis may also be useful. I didn't see any discussion of the outcome of the "Bisection" analyses, so my question is, how did these analyses improve the study's analysis and major findings?. This material may even weaken the study, if it shows changes in response variables preceeded changes to Otomi NPP being switched off. This content comes back to the question in my previous review, was the Otomi NPP being wound down well before the 2012 Otomi survey?, such that an effect could be seen in the 2012 survey at Otomi (for temperature and fish assemblages), or were the 2011/2012 effects at Otomi also partly seen at other sites, and therefore due to a more general environmental effect?. This content then should be more clearly discussed if retained in the manuscript.

If the Bisection analyses don't improve clarity toward resolving this issue of Otomi possibly being wound down prior to the 2012 survey, and details on the Otomi NPP power output (prior to the 2012 survey times) can't be accessed, should the "Bisection" analyses be retained in the manuscript?. If not, the data could just be analysed "as is" (despite the 2012 Otomi on/off uncertainty), strong effects in the study will still be demonstrated.

Response: After carefully considering comments by both reviewers I have deleted the bisection method analyses. 

Presentation

Table 1.

Contains parameter details for site and "year2" (~NPP operating time intervals?). Some of this parameter information is also in Figure 4, so is presented in both "figures and table" (usually not acceptable), and so should be in one or the other. Table 1 does additionally contain p-value information for F and t tests of significance between fixed effect levels (ie. with site and "year2"), but these are poorly regarded (outlined above), and presents parameter estimates in a way that can be confusing to people not familiar with this statistical model format. The F and t tests could be dropped, and Chi-square tests for simply differences between "sites" could be presented instead. Parameter values for each site are then in an updated Fig. 4 (suggestion below), instead of in Table 1. Tests for differences between Otomi time periods (on/off) are not needed in models, if analyses for Otomi "on" and Otomi "off" are conducted separately (and reported separately within Table 1, with site estimates presented in Fig 4 (suggestion)). This reporting for each variable in the current Table 1, could be extended to the Tropical, Sub-tropical and Temperate species subdivisions within the Abundance and Richness variables. Tests for these subdivisions currently appear obscurely as labels in Figure 4, and introduce some clutter into the figure.

Response: Table 1 has also been deleted in the revised manuscript, as it was accompanied with the bisection method model analyses and GLMMs.

Figure 2.

This Figure contains un-necessary information in the form of standard errors for each year*site data point. With the dropping of the 2-factor ANOVA approach to analysis, this information is no longer really needed, and would instead be provided more directly in an updated Figure 4 (see below). However, this figure can still provide useful information, in depicting how the raw data varies over sites. For the case of the temperature data, adding lines to adjacent year means, gives a more clearly digestable idea of how sites can be quite consistently ranked for mean temperature over the series of years that were studied, and showing strongly the effect of the heat source being cut at Otomi around 2012 (and perhaps the same shown for "fish abundance", etc...).

So perhaps confidence intervals can be dropped from Fig. 2, and lines added linking adjacent year means for each site. This altered figure is especially useful for temperature, as this is driving the biological effects in the study.

Response: I thank the reviewer again for this constructive comment. Data presentation has been changed to provide row data plot and year means drawn as lines for each site as was suggested in the revised Fig 2. 

Figure 3.

discussed above, still needed?

Response: The Figure has been deleted together with the bisection method model analyses.

Figure 4.

Error bars presented in Figure 4 of the previous manuscript, were changed from standard deviations to standard errors. This is an improvement but still not optimal. I have included in an attachment, some R-code on how to calculate 95% confidence intervals for fixed effects from mixed models, by "simulating" mixed model fits (from lmer(model) ), and then extracting 0.025 and 0.975 percentiles. To illustrate, this was done for the temperaure data now included in Table S1, for when Otomi NPP was operating, and when it wasn't. The confidence intervals have been plotted in an attached Figure, generated by R (code attached). I recommend using 95% CI's, because significant differences can be noted directly from these plots, eliminating the need to provide information currently in Table 1. The author might need to reconsider grouping separate plots in Fig. 4 around variables (for greatest test validity), rather than sites, as is currently the case. Temperature confidence intervals could be included in Figure 4, or a separate small similarly structured Figure.

A rule of thumb applies to presentations comparing treatment means via 95% confidence intervals. Where 2 intervals just fail to overlap, the treatment levels differ at approximately p=0.01. Confidence intervals will differ at p~0.05 if the overlap is about a third of one side of an an error bar (Cumming (2012) ).

Please note that Figure 4 should plot both upper and lower confidence intervals, to enable quick visual comparisons to be made between "sites"/"treatments". Bar plots can be used to plot estimated means, or points, as in the temperature example provided, but any bar shading should obviously be light enough not to obscure the lower bars.

The author plotted standard errors in his new Figure 4, because his prefered plotting software (Kaleida Graph) couldn't draw 95% CI's. If Kaleida Graph does not have an option to plot user supplied errors (where this can't be directly generated within the software), other plotting software to do this is available. R can be programmed to produce such plots (see attached document for code), though time consuming to implement.

Response: I sincerely thank the reviewer for providing the R code. I applied this procedure to each response variable, resulting in the new Figure 3 and revised Figure 4. Means and confidence intervals were obtained through the R codes with simulation, then graphs were drawn by Kaleida Graph with these data.

---

## [Editor Report · Decision Letter 2]

2 Apr 2020

PONE-D-19-30294R2

Tropical fishes vanished after the operation of a nuclear power plant was suspended in the Sea of Japan

PLOS ONE

Dear Dr. Masuda,

Thank you for submitting your manuscript to PLOS ONE. After careful consideration, we feel that it has merit but does not fully meet PLOS ONE’s publication criteria as it currently stands. Therefore, we invite you to submit a revised version of the manuscript that addresses the points raised during the review process.

I think the author has done an excellent job taking the statistical advice provided on the previous version of this manuscript. The paper is much clearer and more concise and the presentation of the results has definitely improved. The author has introduced some grammatical errors and there are still a few bits that are unclear so I think that needs to be corrected before the manuscript can be accepted. However, these corrections (PLoS editorial comment file) will not take long.

We would appreciate receiving your revised manuscript by May 17 2020 11:59PM. To enhance the reproducibility of your results, we recommend that if applicable you deposit your laboratory protocols in protocols.io, where a protocol can be assigned its own identifier (DOI) such that it can be cited independently in the future. For instructions see: http://journals.plos.org/plosone/s/submission-guidelines#loc-laboratory-protocols

We look forward to receiving your revised manuscript.

Kind regards,

Heather M. Patterson, Ph.D.

Academic Editor

PLOS ONE

---

## [Author Response · Author response to Decision Letter 2]

5 Apr 2020

Responses to the Editor

Line 42: Should be ‘districts’

Line 50: Delete ‘the’ before ‘extension’

Response: I have corrected the above two points according to the suggestion.

Lines 136-139: I find this text a little confusing as written. Maybe rewrite as something like:

‘The center of distribution (COD), defined as the mean southern and northern latitudinal limit of distribution in the northern hemisphere, was calculated using Nakabo [19] for all recorded species. The COD was then compared to the mean latitude recorded in each survey and used as a criteria of spatio-temporal tropicalization.’

Does that work? That seems to be what has been done but it is not entirely clear from the text as it is written.

Response: I thank the editor for notifying this ambiguity. I have made revisions on these sentences as follows (L136-139):

The center of distribution (COD), defined as the mean southern and northern latitudinal limit of distribution in the northern hemisphere, was calculated using Nakabo [19] for all recorded species. The mean of COD in each survey was then used as a criteria of spatio-temporal tropicalization.

Line 14: delete ‘of’ before ‘lme4’

Response: I corrected it according to the suggestion.

Line 145: provide the years the NPP was in operations in parentheses after ‘operation’

Line 146: Time period confusing as written. Should be simplified to something like ‘after 20 february 2012’ or something similar

Response: I have revised this part as following (L 144-146):

The analyses were conducted separately in two periods when NPP was in operation (from 2004 to early February 2012) or suspended (February 20, 2012 and later).

Line 149: What does ‘above category’ mean>

Response: I have clarified this phrase as follows (L 149-150):

above category (tropical, subtropical and temperate species)

Line 151L Write as ‘by simulating linear mixed model fits (10,000 runs)

Line 152: ‘significant’ is misspelled

Line 153: Need a comma before ‘as well’

Line 178: Write as ‘Lines represent yearly means for each site (surveys = 4)’ or something like that

Line 182: Shouldn’t this be ‘and later’?

Lines 184-185: Write as ‘Error bars represent 95% confidence intervals.’

Line 204: Should be ‘higher tropical fish abundance’

Line 208: Should be ‘Mean temporal changes’

Lines 210-211: Write as ‘Error bars represent 95% confidence intervals.’

Line 217: Delete ‘either’

Line 229: Should be ‘the NPP’ and need a comma after ‘suspension’

Line 264: write as ‘and therefore the small increase’

Line 283: Should be ‘effects’

Line 295: I assume this means that larval fish spawned in this area likely self-recruited so should be something like ‘Locally spawned larval fish of these species may have…..’

Response: Corrections have been made according to the above suggestions.

---

## [Editor Report · Decision Letter 3]

7 Apr 2020

Tropical fishes vanished after the operation of a nuclear power plant was suspended in the Sea of Japan

PONE-D-19-30294R3

Dear Dr. Masuda,

We are pleased to inform you that your manuscript has been judged scientifically suitable for publication and will be formally accepted for publication once it complies with all outstanding technical requirements.

With kind regards,

Heather M. Patterson, Ph.D.

Academic Editor

PLOS ONE
---

## [Editor Report · Acceptance letter]

9 Apr 2020

PONE-D-19-30294R3 

Tropical fishes vanished after the operation of a nuclear power plant was suspended in the Sea of Japan 

Dear Dr. Masuda:

I am pleased to inform you that your manuscript has been deemed suitable for publication in PLOS ONE. Congratulations! Your manuscript is now with our production department. 

With kind regards,

on behalf of

Dr. Heather M. Patterson 

Academic Editor

PLOS ONE